# Nomogram Predicting In-Hospital Mortality in Patients with Myocardial Infarction Treated with Primary Coronary Interventions Based on Logistic and Angiographic Predictors

**DOI:** 10.3390/biomedicines13030646

**Published:** 2025-03-06

**Authors:** Lukasz Gawinski, Anna Milewska, Michal Marczak, Remigiusz Kozlowski

**Affiliations:** 1Department of Management and Logistics in Healthcare, Medical University of Lodz, 90-419 Lodz, Poland; lgaw@amg.gda.pl; 2Department of Cardiology, Invasive Cardiology and Electrophysiology with Intensive Cardiac Care Subunit, Regional Specialist Hospital, 86-300 Grudziadz, Poland; 3Department of Biostatistics and Medical Informatics, Medical University of Bialystok, 15-295 Białystok, Poland; anna.milewska@umb.edu.pl; 4Department of Innovation, Merito University in Poznan, 03-204 Warszawa, Poland; michal.j.marczak@gmail.com

**Keywords:** myocardial infarction, primary coronary intervention, risk score

## Abstract

**Background:** Systems developed in recent years to assess the risk of in-hospital death in patients with myocardial infarction (MI) are mainly based on angiographic, electrocardiographic, and laboratory variables. Risk systems based on contemporary angiographic data and logistic variables have not been reported. The aim of this study was to develop and validate a system to assess the risk of in-hospital death in patients across the entire clinical spectrum of MI treated with primary coronary intervention (PCI) based on modern angiographic and logistic predictors. **Methods:** A subgroup of patients from the observational single-centre registry of MI treated with PCIs (from 1 February 2019 until 31 January 2020) was used to develop a multivariate logistic regression model predicting in-hospital mortality. The population (603 patients) was divided, with 60% of the sample used for model derivation and the remaining 40% used for internal model validation. **Results:** The main findings were as follows: (1) coronary angiography results and suboptimal flow after PCI were important predictors of in-hospital mortality; (2) the time of PCI as well as the mode of presentation of patients with MI contributed to in-hospital mortality; and (3) the discrimination (C statistic = 0.848, 95% CI: [0.765, 0.857]) and calibration (χ^2^ = 2.78, pHL = 0.94) were good in the derivation set, while the discrimination (C statistic = 0.6438, 95% CI: [0.580, 0.703]) in the validation set was satisfactory. **Conclusions:** A novel clinical nomogram based on four available logistic and angiographic variables was developed and validated for in-hospital mortality after PCIs in a wide range of MIs.

## 1. Introduction

The widespread use of invasive treatments for myocardial infarction (MI) in the 21st century has become the standard of care to ensure the best outcomes. Although the history of invasive treatments for MI goes back several decades, recent years have seen further declines in both short- and long-term mortality rates [1]. Several factors have been implicated in this phenomenon. One of the most important is the rapid technological development of invasive treatment methods associated with the use of new materials and devices and the widespread use of new-generation drug-eluting stents (DESs) [2]. Other important factors include the routine introduction of new antiplatelet drugs [3] and the development of local MI treatment networks—consisting of a system of catheterisation laboratories (CLs) and emergency medical services (EMS) teams—supported by increasingly advanced and efficient telematics systems [4]. Over the past few decades, a number of systems have been developed to assess the risk of death in patients with MI. The first risk assessment systems mainly focused on clinical factors, laboratory findings, and electrocardiographic presentation [5]. With the gradual spread of invasive treatments for MI, angiographic predictors have been increasingly incorporated into risk assessment systems. Systems published in recent years to assess the risk of in-hospital death in patients with MI either do not include angiographic factors related to invasive procedures for MI (Grace Score [6], ProACS Score [7], and ACTION-GWTG Score [8]) or include angiographic factors in their schemes based on clinical data from 2004 to 2008 (AR-G Score [9], EH STEMI PCI Score [10], and NCDR Cath PCI Score [11]), making them outdated. Other limitations of previously published systems for assessing the risk of in-hospital death in patients with MI include the focus on the ST-elevation myocardial infarction (STEMI) patient population—without including patients with non-ST-elevation myocardial infarction (NSTEMI) or unstable angina (UA) [12,13]—and the heterogeneity of the study population (patients were treated both invasively and pharmacologically with fibrinolytic therapy) [6,9]. Recent reports suggest that the logistic aspects of the management of MI may also significantly influence a patient’s risk of in-hospital death [14]. To the best of the authors’ knowledge, none of the existing systems for assessing the risk of in-hospital death in patients with MI include logistic aspects in their algorithms [15]. These facts make it necessary to update the systems for assessing the risk of in-hospital death in patients with MI based on the latest clinical data on primary coronary interventions (PCIs), taking into account broad logistical aspects. The aim of this study was to develop and validate a new, innovative system for assessing the risk of in-hospital death in patients with full spectrum of MI (STEMI and NSTEMI/UA) treated with PCIs based on angiographic and logistic predictors: the ANGIOLOG IH Risk Score (ANGIOgraphic–LOGigistics In-Hospital Death Risk Score).

## 2. Materials and Methods

### 2.1. ACS GRU Registry

The Registry of Acute Coronary Syndromes is a single-centre, retrospective, observational registry of patients admitted with MI to the Department of Cardiology, Invasive Cardiology and Electrophysiology with Intensive Cardiac Care Subunit (hereafter called the Department of Cardiology) of the Regional Specialist Hospital (RSH) in Grudziadz (Poland) (ACS GRU registry). The design and methods of the ACS GRU registry have been described previously [14]. Medical data were included in the registry based on retrospective analysis of electronic medical records without direct patient contact. The only criterion for inclusion in the registry was the diagnosis of MI in an adult patient in accordance with the current European Society of Cardiology (ESC) guidelines for the diagnosis of MI [16]. In general, the registry had no exclusion criteria. The patients included in the registry presented with the full spectrum of MI: STEMI, NSTEMI, and UA. The registry included patients treated both invasively and conservatively. Finally, the registry included 633 adult patients (35.39% of whom were women) with an average age of 67.95 years (±4.95) admitted to the Department of Cardiology of the RSH in Grudziadz for MI during 12 consecutive months (from 01.02.2019 to 31.01.2020). All patients were of European descent. The patients’ personal data were retrospectively anonymised and entered into the electronic ACS GRU registry.

### 2.2. Study Population

A group of patients treated with PCIs were identified within the main study group (*N* = 603; 33.5% female; 35.99% STEMI; mean age: 67.37 years (±11,096)). None of the patients enrolled in this study had previously received thrombolytic treatments. The study population was randomised into 2 sets: (1) a derivation set (*n* = 359 subjects) representing 60% of the study population and (2) an internal validation set (*n* = 244 subjects) (40%). Details of the final size of each set are shown in Figure 1. Information on missing data is provided in Appendix A. Patient cases with missing data were not included in further analyses, and no imputation techniques were used. The characteristics of the two sets are shown in Table 1. The characteristics of the final two sets (after exclusion of patients with MI with non-obstructive coronary arteries, patients referred for coronary artery bypass grafting, patients treated conservatively after coronarography, and missing data) are shown in in Appendix A. Study was conducted with the approval of the Local Bioethics Committee (No. 8/KB/2021). It was designed in accordance with the tenets of the Declaration of Helsinki [17] and good clinical practice.

### 2.3. Selection of Variables

For the purpose of this model, all variables available in the ACS GRU registry related to the invasive treatment parameters and logistical aspects of the MI treatment process were selected according to the original assumptions. The variables ultimately selected for analysis were as follows:A panel of general variables: sex, patient age, and STEMI/NSTEMI/UA;A panel of angiographic variables: vascular access, extent of coronary atherosclerotic lesions—result of coronary angiography (CAG), PCI on the left main or proximal left anterior descending artery (PCI LM/prox LAD), restenosis in a DES in the infarct-related artery (IRA), Thrombolysis in Myocardial Infarction (TIMI) flow grade before and after PCI, pre-dilatation with balloon (semi-compliant, SC; or non-compliant, NC), post-dilatation with NC balloon after stent implantation, bifurcation PCI, coronary artery calcification in IRA, cardiopulmonary resuscitation (CPR) during PCI, and unsuccessful PCI;A panel of variables related to the logistical aspects of the MI treatment process: type of presentation, time of hospital admission, and time of PCI.

### 2.4. Endpoint and Definitions

The endpoint was in-hospital death of a patient with MI, regardless of the cause or site (CL, intensive care unit, or cardiac surgery unit). The treatment strategies for the patients with MI, including the individual procedures and pharmacotherapies used, are included in the description of the ACS GRU registry [14]. Right radial artery access was used as a standard; in the case of failure, the operator personally selected another type of vascular access (left radial artery, right or left femoral artery, or another artery). The result of CAG was determined according to the following criteria: coronary arteries without significant stenosis, single-vessel disease, double-vessel disease, multi-vessel disease with involved LM, and multi-vessel disease without involved LM. The degree of coronary flow was determined according to the TIMI flow grade [18]. The bifurcation lesion was defined as the side branch requiring wire protection from the culprit lesion. Angiographic coronary artery calcification was assessed by the operator. Calcifications were defined as radiographic changes that were visible on the coronary view prior to the administration of contrast media [19]. For a PCI to be considered unsuccessful, the following criteria had to be met: culprit lesion not crossable using a wire or inability to cross the lesion with a balloon or stent. Patients with MI could present to the Department of Cardiology in three ways:Admitted directly from home/a public place (patient was transferred by the EMS team or arrived at the Emergency Department on their own);Transferred from another hospital;Transferred from another department of the parent hospital to which they were previously admitted for another condition.

The analysis of the logistical aspects led to the classification of the time of admission to hospital/time of PCI into weekdays and holidays. In the study centre, a regular working day runs from 8:00 to 14:00 on weekdays. The following subperiods were selected: from 8:00 to 14:00; from 14:00 to 22:00 (daytime duty); and from 22:00 to 8:00 of the following day (night-time duty). On public holidays, the following subperiods were selected: from 8:00 to 22:00 (daytime duty) and from 22:00 to 8:00 of the following day (night-time duty).

### 2.5. Statistical Analyses

Categorical variables were presented as frequencies and percentages. Differences in mortality in specific subgroups of patients were assessed using the chi-squared test. Quantitative variables were transformed into categorical variables; patient age was presented in the form of 10-year groups, and the TIMI flow grade was presented in the form of 2 groups: group 1 (TIMI flow grades 0–1) and group 2 (TIMI flow grades 2–3). In the first step, a univariate logistic regression model was applied to assess the associations between each predictor variable and the outcome of in-hospital death in the derivation set. In the next step, a multivariate model was obtained using the backward stepwise strategy. A nomogram is a medical tool that graphically calculates the risk of a particular outcome for an individual. These useful and simple bedside prediction tools assign scores to each value level of each variable (according to the degree of contribution of each variable in a model). The total score is obtained by summing all the scores. The regression coefficient estimates for each significant variable (*p* < 0.05) were converted into integers that allowed for a mortality odds score to be calculated for each patient. The area under the receiver operating characteristic curve was used to assess the discriminative performance of the predictive nomogram [20]. The predictive accuracy of the model was evaluated using the Hosmer–Lemeshow test [21]. Graphically, this test is presented using a calibration curve. The performance of the model was assessed in terms of discrimination in the validation set using the same method described above. Decision curve analysis (DCA) was performed to assess the clinical utility of the nomogram [22]. *p* values < 0.05 were considered statistically significant. The data are presented as odds ratios (ORs) with 95% confidence intervals (95% CIs). The statistical analysis was conducted utilising Stata ver. 17.

## 3. Results

In terms of baseline characteristics, no important differences were observed between the derivation (*n* = 362) and validation (*n* = 244) sets. Random allocation between the derivation and validation sets was demonstrated by *p* values < 0.05 (Appendix A). The all-cause mortality in the study population was 5.55%. There was no statistically significant difference between the in-hospital mortality rates of the patients with MI in the individual sets, which were 5.57% and 6.15% (*p* = 0.776) in the derivation and validation sets, respectively. The results of the univariate and multivariate logistic regression analysis in the derivation set are presented in Table 2. Based on the multivariable logistic regression model, the following variables were selected as predictors for the development of the mortality odds prediction model: the mode of presentation, the result of CAG, the post-PCI TIMI flow grade, and the time of PCI. The mortality score was calculated as follows:Mortality score = exp (MP2 × 0.8041279 + MP3 × 2.211911 + RCAG2 × 0.0568784 + RCAG3 × 0.9143116 + RCAG4 × 2.038374 + post-PCI TIMI × 2.286594 + TPCI2 × 1.149752 +TPCI3 × 1.897038 + TPCI4 × 0.3486303 − 4.837896)
including the mode of presentation type 2 (MP2; patients admitted from another hospital), the mode of presentation type 3 (MP3; patients admitted from another department), the result of CAG type 2 (RCAG2; double-vessel disease), the result of CAG type 3 (RCAG3; multi-vessel disease with affected LM), the result of CAG type 4 (RCAG4; multi-vessel disease without affected LM), the time of PCI type 2 (TPCI 2; weekday from 14:00 to 22:00), the time of PCI type 3 (TPCI 3; weekday from 22:00 to 8:00), and the time of PCI type 4 (TPCI 4; public holiday from 8:00 to 22:00). The post-PCI TIMI flow grades (post-PCI TIMI) 0–1 were scored with yes as 1 and no as 0. This equation was used to construct a nomogram to predict the probability of in-hospital death for the patients with MI in the derivation set (Table 3, Figure 2). Each variable was assigned a score on a scale. These scores were then added to obtain the total score, and a vertical line was drawn from the total score line to estimate the probability of in-hospital death. Discrimination (C statistic = 0.848, 95% CI: [0.765, 0.857]) and calibration (χ^2^ = 2.78, pHL = 0.94) were very good in the derivation set (Figure 3a and Figure 4). The discrimination (C statistic = 0.6438, 95% CI: [0.580, 0.703]) in the validation set was satisfactory (Figure 3b). The decision curve analysis of the nomogram model is shown in Figure 5. When the predicted risk of in-hospital mortality in patients with MI after PCI was 0.01–0.81, more significant net benefits were achieved when implementing treatment measures than when not applying treatments.

## 4. Discussion

This study presents a novel risk score for predicting in-hospital mortality after PCI across a wide range of MIs. The score derived for in-hospital mortality included four readily available variables and, for the first time, logistic parameters. It is also noteworthy that pre- and post-procedural angiographic variables were combined in this model. The main findings were as follows:Angiographic variables, such as CAG results and suboptimal flow after PCI, were important predictors of in-hospital mortality;Logistical aspects of the MI treatment process, such as the time of PCI and the mode of presentation of patients with MI, contributed to in-hospital mortality;Discrimination was good in both the derivation and validation sets.

The Hosmer–Lemeshow goodness of fit test showed that the deviation between the risk prediction value of the nomogram model and the actual observed value was not statistically significant. This means that the nomogram model had a good fit, and its prediction of the probability of in-hospital mortality after PCI demonstrated good concordance with the actual probability. As the GRACE scale has been the standard for assessing mortality in patients with MI for many years, it is good practice to compare any new risk scale with the GRACE scale. Depending on the version, the GRACE score is based on a set of indicators: clinical, electrocardiographic and laboratory parameters, as well as medical history data. In fact, all subsequent models of the GRACE score are based on the same or similar set of risk factors assessed at hospital admission. GRACE score is currently recommended by the ESC for risk stratification and to identify indications for early invasive diagnosis in patients with NSTEMI (class IIa, the level of evidence :B). The c-statistic of GRACE varied according to the patient subgroup selected and was >0.8. The score was based on the analysis of data from the GRACE registry (11,389 patients with MI, both STEMI and NSTEMI, enrolled between 1 April 1999 and 31 March 2001). The ANGIOLOG IH Scale is constructed from angiographic and logistic variables and has a c-statistic of 0.848. The score was based on the analysis of data from the ACS GRU registry (patients with MI, both STEMI and NSTEMI/UA, enrolled between 2019 and 2020). The GRACE score is assessed at hospital admission. Risk stratification with the ANGIOLOG IH scale is performed after the completion of invasive treatment.

### 4.1. Angiographic Variables

The extent of atherosclerotic lesions detected by CAG in patients with MI is a recognised factor that negatively influenced in-hospital mortality in this group of patients. These results are consistent with the results of other studies on models of the risk of in-hospital death in patients with MI [10]. The vast majority of these studies only included patients with STEMI (which, in most cases, limited the CAG results to single- or double-vessel disease). The discussed risk score also applies to patients with NSTEMI/UA, which results in an increased number of patients with multi-vessel disease with/without affected LM and undoubtedly proves the greater usefulness of this score in everyday clinical practice (taking into account that the number of STEMIs in Europe is currently much lower than the number of NSTEMIs [23]). The significantly lower risk of death in patients with multi-vessel disease without affected LM compared to patients with multi-vessel disease with affected LM may be questionable (this result is not statistically significant). The reason for this may be that patients with multi-vessel disease with affected LM, who would be expected to have the highest risk of death during PCI and later hospitalisation, were often referred for urgent cardiac surgery and were, therefore, not included in the final study population.

The effect of impaired coronary flow after PCI (known as the no-reflow phenomenon) is a well-established predictor of poor prognosis in patients with MI [24] and has been used repeatedly in previously developed risk models [25]. The risk factors for no reflow can be divided into those related to the patient (previous clinical stress, patient age, and nicotine addiction) and those related to the PCI (high-pressure inflation and the use of debulking techniques). The causes of this phenomenon are complex and are related to functional and structural disturbances of the coronary microcirculation [26].

### 4.2. Logistic Variables

The results regarding the impacts of logistic aspects on in-hospital mortality in patients with MI treated with PCIs are much more interesting and innovative. The results of a meta-analysis evaluating the impact of the time of hospital admission on short-term mortality in patients with MI were unclear and contradictory [27]. This meta-analysis included over 49 trials involving patients with MI (both STEMI and NSTEMI, with a predominance of patients in the former group) published between 2000 and 2020. It should also be noted that, in patients with STEMI, the time of hospitalisation was most often the same as the time of PCI (assuming the shortest possible door-to-balloon time), whereas in patients with NSTEMI these two periods could differ significantly for various reasons. Assuming that most of the risk of in-hospital death during the treatment of MI is related to PCI, and taking into account the above considerations, it seems that a much more useful and universal logistic parameter that may influence in-hospital mortality is the time of PCI (especially in the population of patients with NSTEMI). According to the latest ESC guidelines for the management of patients with NSTEMI [28], if an early invasive strategy is chosen, PCI can be performed within 24 h. This allows the time of PCI to be shifted for some patients to the regular hours of operation of the CL which, in turn (given the results of the present study), may significantly minimise the risk of in-hospital death. These findings were confirmed by a paper published in 2021 that analysed the effect of the time of PCI on in-hospital mortality in a group of more than 99,000 patients [29]. Despite the relatively well-defined time intervals for PCI in patients with NSTEMI in the aforementioned ESC guidelines, there has been an ongoing debate in recent years about the appropriate timing of PCI in these patients. Recent reports based on large registries of patients with NSTEMI do not support an early invasive strategy, especially in the group of high-risk patients [30], and point to the need to individualise decisions on the appropriate timing of invasive treatment based on symptoms and the full spectrum of possible risks. The results of the present study are in line with these suggestions.

Another logistic variable included in the nomogram is the mode of presentation: admission from home/a public place, admission from another hospital, or admission from another department of the hospital performing the PCI. The negative impact of the prolonged period from first medical contact to the balloon procedure associated with the inter-hospital transfer of a patient with MI has been well described and documented in the literature [31]. According to the results of the present analysis, patients transferred to the study centre from another department in the same hospital (where they were admitted for a condition other than MI) who received a diagnosis of MI during their hospitalisation had a significantly higher risk of in-hospital death than patients transferred from home/a public place. These associations have been observed for many years, particularly in patients in intensive care units [32].

### 4.3. Limitations

This study had several limitations:Considering the retrospective nature of this study, some unknown factors were prone to data deviation and led to inevitable bias;The present investigation was a single-centre study with a small sample size. The risk factors included in this study were not comprehensive, and bias could not be avoided;In terms of model validation, only internal validation was performed;Due to the small size of the population (and the random allocation of the patients to the derivation and validation sets), there were no patients in the derivation set who underwent PCI on the weekend between 22:00 and 8:00.

## 5. Conclusions

In conclusion, the present score innovatively combines angiographic and logistic variables in the assessment of in-hospital mortality in patients after PCIs in a wide range of MIs. It demonstrated good discrimination, calibration, and high clinical utility and versatility. This nomogram will provide clinicians with a rapidly accessible and reliable clinical scoring tool for the early clinical prediction of the risk of in-hospital death after PCI in patients with MI, as well as for the identification of high-risk patients in whom close monitoring and other interventions may be warranted. The developed risk assessment model provides more interdisciplinary and realistic insight into the assessment of the short-term prognosis of patients with MI treated with PCIs. Logistic factors, unlike other classical clinical variables, can be actively modified in specific cases, and so the described risk model also allows a certain degree of active management of the risk of an in-hospital death of patients with MI. The developed risk model works in a multifaceted way, both in terms of the logistic criteria (pre-hospital and in-hospital periods) and angiographic criteria (angiographic outcome and final PCI effect), thus increasing the cardiologist’s clinical awareness during the clinical decision-making process. It allows primary (preoperative) and secondary (postoperative) assessments of the risk of mortality. To the best of the authors’ knowledge, this is the first model to assess the risk of in-hospital death in patients with MI based on logistic variables.

## Figures and Tables

**Figure 1 biomedicines-13-00646-f001:**
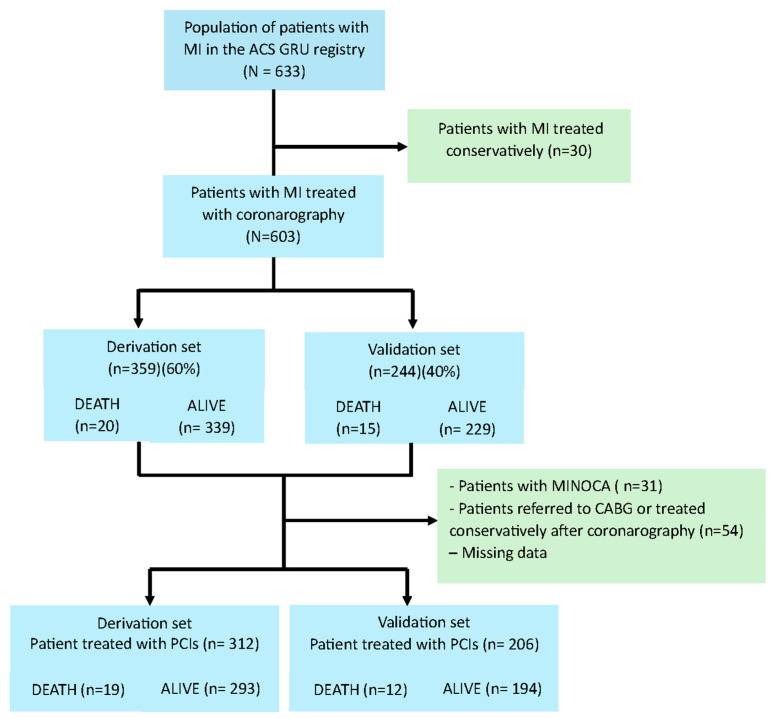
Patient flow diagram. The study population was from the single-centre ACS GRU registry (included a total of 663 patients). The final study group consisted of patients with MI who were treated with PCIs and had complete clinical data (*N* = 516). CABG—coronary artery bypass grafting, MI—myocardial infarction, MINOCA—myocardial infarction with non-obstructive coronary arteries, and PCI—primary coronary intervention.

**Figure 2 biomedicines-13-00646-f002:**
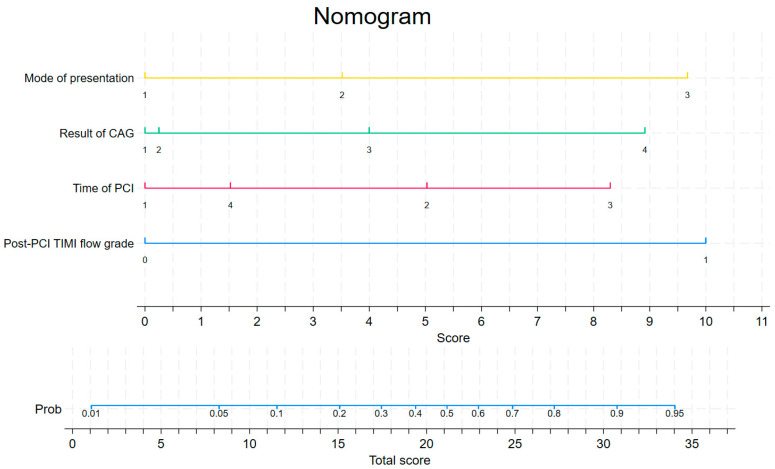
Nomogram predicting in-hospital mortality in patients with myocardial infarction treated with primary coronary interventions based on logistic and angiographic predictors.

**Figure 3 biomedicines-13-00646-f003:**
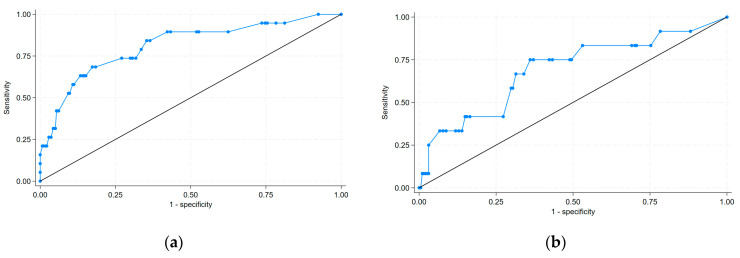
Discrimination of the in-hospital mortality score in: (**a**) derivation set; (**b**) validation set. ROC curve of the nomogram in the derivation set (**a**) and validation set (**b**). ROC = receiver operating characteristic.

**Figure 4 biomedicines-13-00646-f004:**
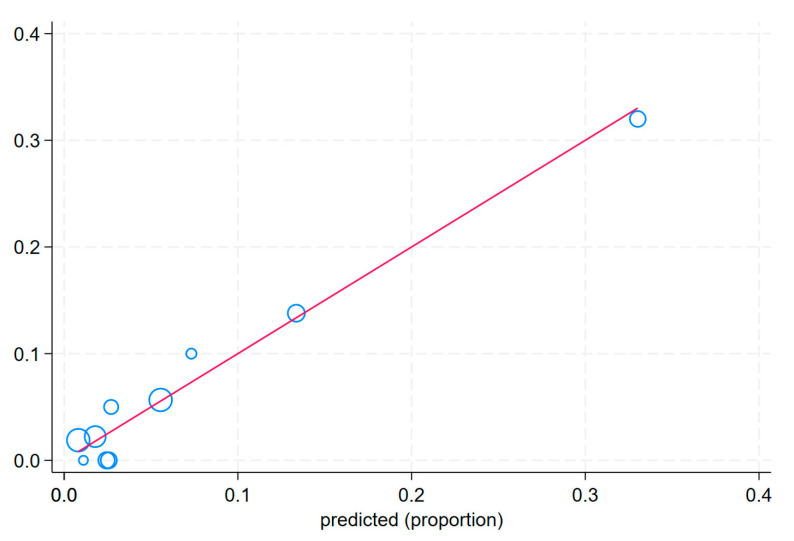
Calibration curves for the in-hospital mortality score in the derivation set. The Hosmer–Lemeshow test (χ^2^ = 2.78, pHL= 0.94) was used for predictive accuracy of the model. Graphically, this test is presented using a calibration curve. Blue circle—observed, red line—predicted.

**Figure 5 biomedicines-13-00646-f005:**
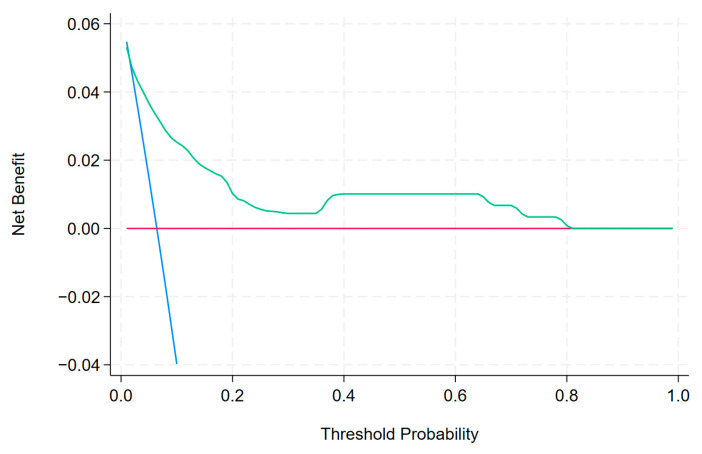
The decision curve analysis of the nomogram model. The benefits observed in the treatment group were more significant than those seen in the group without treatment when the predicted risk of death in patients MI was found to range from 0.01 to 0.81. The decision curve analysis shows that the nomogram can achieve a good net benefit. Red line—none, blue line—all, green line—nomogram.

**Table 1 biomedicines-13-00646-t001:** Clinical characteristics of the patients in the derivation and validation sets according to their in-hospital mortality status.

Variables	Derivation Set (*n* = 359)		Validation Set (*n* = 244)	
	Dead (*n* = 20)	Alive (*n* = 339)	Dead (*n* = 15)	Alive (*n* = 229)
General factors:				
Age (per 10-year increase)				
<40 years old	20.00% (1)	80.00% (4)	0% (0)	100% (3)
40–49 years old	9.09% (1)	90.91% (10)	0% (0)	100% (9)
50–59 years old	0% (0)	100% (72)	8.70% (4)	91.30% (42)
60–69 years old	4.62% (6)	95.38% (124)	3.30% (3)	96.70% (88)
70–79 years old	6.74% (6)	93.26% (83)	8.00% (4)	92.00% (46)
89–89 years old	10.64% (5)	89.36% (42)	6.98% (3)	93.02% (40)
≥90 years old	20.00% (1)	80.00% (4)	50.00% (1)	50.00% (1)
Women	3.42% (4)	96.58% (113)	9.41% (8)	90.59% (77)
Men	6.61% (16)	93.39% (226)	4.40% (7)	95.60% (152)
STEMI	8.06% (10)	91.94% (114)	8.60% (8)	91.40% (85)
NSTEMI/UA	4.26% (10)	95.74% (225)	4.64% (7)	95.36% (144)
Angiographic factors:				
Vascular access:				
Right radial artery	4.03% (12)	95.97% (286)	4.31% (9)	95.69% (200)
Left radial artery	13.79% (8)	86.21% (50)	18.18% (6)	81.82% (27)
Right/left femoral artery	0% (0)	100% (3)	0% (0)	100% (2)
Result of coronarography study:				
One-vessel disease	2.33% (2)	97.67% (84)	6.90% (4)	93.10% (54)
Double-vessel disease	3.26% (3)	96.74% (89)	1.92% (1)	98.08% (51)
Multi-vessel disease with affected LM	6.56% (8)	93.44% (114)	3.41% (3)	96.59% (85)
Multi-vessel disease without affected LM	13.95% (6)	86.05% (37)	16.13% (5)	83.87% (26)
PCI on LM/proximal LAD	11.90% (10)	86.10% (74)	9.1% (4)	90.9% (40)
Restenosis in a DES in IRA	6.25% (1)	93.75% (15)	16.67% (1)	83.33% (5)
Post-PCI TIMI flow grades 0–1	44.44% (4)	55.56% (5)	50.00% (2)	50.00% (2)
Post-PCI TIMI flow grades 2–3	4.95% (15)	95.05% (288)	4.95% (10)	95.05% (192)
Pre-dilatation with balloon (SC or NC)	6.15% (16)	93.85% (244)	5.75% (10)	94.25% (164)
Post-dilatation with NC balloon	3.69% (8)	96.31% (209)	3.95% (6)	96.05% (146)
PCI in bifurcation	2.38% (1)	97.62% (41)	4.00% (1)	96.00% (24)
Coronary artery calcifications in IRA	12.5% (6)	87.5% (42)	6.67% (2)	93.33% (28)
CA with subsequent CPR (CL stage)	50.00% (4)	50.00% (4)	50.00% (2)	50.00% (2)
Unsuccessful PCI	50.00% (4)	50.00% (4)	50.00% (2)	50.00% (2)
Logistical factors:				
Mode of presentation				
Admission from home/public place	4.07% (12)	95.93% (283)	6.73% (14)	93.27% (194)
Admission from another hospital	8.33% (4)	91.67% (44)	0.00% (0)	100% (28)
Admission from another department	25.00% (4)	75.00% (12)	12.50% (1)	87.50% (7)
Time of hospital admission:				
Weekday from 8:00 to 14:00	2.21% (3)	97.79% (133)	3.49% (3)	96.51% (83)
Weekday from 14:00 to 22:00	11.96% (11)	88.04% (81)	5.56% (3)	94.44% (51)
Weekday from 22:00 to 8:00	6.82% (3)	93.18% (41)	7.69% (3)	92.31% (36)
Public holiday from 8:00 to 22:00	3.57% (2)	96.43% (54)	2.33% (1)	97.67% (42)
Public holiday from 22:00 to 8:00	3.23% (1)	96.77% (30)	22.73% (5)	77.27% (17)
Time of PCI:				
Weekday from 8:00 to 14:00	1.59% (2)	98.41% (124)	5.49% (5)	94.51% (86)
Weekday from 14:00 to 22:00	8.40% (11)	91.60% (120)	4.17% (3)	95.83% (69)
Weekday from 22:00 to 8:00	14.29% (4)	85.71% (24)	15.00% (3)	85.00% (17)
Public holiday from 8:00 to 22:00	5.26% (3)	94.74% (54)	4.55% (2)	95.45% (42)
Public holiday from 22:00 to 8:00	0% (0)	100% (17)	11.76% (2)	88.24% (15)

Values are percentages (n). CA—cardiac arrest; CL—catheterisation laboratory; CPR—cardiopulmonary resuscitation; IRA—infarction-related artery; LAD—left anterior descending artery; LM—left main;; NC—non-compliant; PCI—primary coronary intervention; SC—semi-compliant; STEMI—ST-elevation myocardial infarction; TIMI—Thrombolysis in Myocardial Infarction flow grade.

**Table 2 biomedicines-13-00646-t002:** Results of uni- and multivariate logistic regression analysis in derivation set.

	Univariate Analysis			Multivariate Analysis		
Variables	OR	95% CI	*p* Value	OR	95% CI	*p* Value
General factors:						
STEMI	1.519	0.60–3.85	0.379			
Women	0.52	0.17–1.62	0.260			
Age (per 10-year increase)	1.50	0.99–2.28	0.05			
Angiographic factors:						
Vascular access:						
Right radial artery (ref)						
Left radial artery	1.50	0.99–2.28	0.055			
Right/left femoral artery	/	/	/			
Result of coronarography study:						
One-vessel disease (ref)						
Double-vessel disease	1.37	0.22–8.46	0.729	1.06	0.16–7.15	0.953
Multi-vessel disease with affected LM	3.22	0.66–15.62	0.146	2.49	0.48–13.08	0.279
Multi-vessel disease without affected LM	7.90	1.51–41.32	0.014	7.68	1.34–43.93	0.022
PCI on LM/proximal LAD	3.33	1.30–8.52	0.012			
Restenosis in a DES in IRA	1.04	0.13–8.33	0.970			
Post-PCI TIMI flow grades 2–3 (ref)						
Post-PCI TIMI flow grades 0–1	15.36	3.73– 63.13	0.000	9.84	1.93–50.19	0.006
Pre-dilatation with balloon (SC or NC)	3.22	0.42–24.90	0.261			
Post-dilatation with NC balloon	0.36	0.13–0.96	0.042			
PCI in bifurcation	0.34	0.04–2.62	0.300			
Coronary artery calcifications in IRA	2.76	0.99–7.66	0.051			
CA with subsequent CPR (CL stage)	13.55	2.79–65.72	0.001			
Unsuccessful PCI	19.27	4.39–84.63	0.000			
Logistical factors:						
Mode of presentation:						
Admission from home/public place (ref)						
Admission from another hospital	2.15	0.66–7.04	0.202	2.23	0.63–7.98	0.216
Admission from another department	6.83	1.64–28.54	0.008	9.13	1.75–47.52	0.009
Time of hospital admission:						
Weekday from 8:00 to 14:00						
Weekday from 14:00 to 22:00	5.185	1.38–19.48	0.015			
Weekday from 22:00 to 8:00	3.29	0.63–17.08	0.156			
Public holiday from 8:00 to 22:00	1.55	0.25–9.61	0.634			
Public holiday from 22:00 to 8:00	1.38	0.14–13.81	0.783			
Time of PCI:						
Weekday from 8:00 to 14:00						
Weekday from 14:00 to 22:00	4.90	1.05–22.93	0.043	3.16	0.63–15.83	0.162
Weekday from 22:00 to 8:00	9.18	1.58–53.31	0.013	6.67	1.04–42.63	0.045
Public holiday from 8:00 to 22:00	2.97	0.48–18.34	0.241	1.42	0.20–10.14	0.728
Public holiday from 22:00 to 8:00	/	/	/	/	/	/

Values are odds ratios [95% confidence intervals] for in-hospital mortality in patients with MI treated with PCIs. In instances where no explicit indication to the contrary is provided, the reference group consists of patients for whom the specific variable in question remains unconfirmed. Abbreviations are defined in Table 1.

**Table 3 biomedicines-13-00646-t003:** Risk score for predicting in-hospital mortality in patients with MI.

Mode of Presentation	Score
from home/public place	0
from another hospital	3.5
from another department	9.7
Result of coronarography study	Score
one-vessel disease	0
double-vessel disease	0.2
multi-vessel disease with affected LM	4.0
multi-vessel disease without affected LM	8.9
Time of PCI	Score
weekday from 8:00 to 14:00	0
weekday from 14:00 to 22:00	5
weekday from 22:00 to 8:00	8.3
public holiday from 8:00 to 22:00	1.5
Post-PCI TIMI flow grade	Score
TIMI flow grades 0–1	10.0
TIMI flow grades 2–3	0

Numerical values of individual variables in the risk scale for the assessment of in-hospital mortality in patients with MI treated with PCIs. Abbreviations are defined in Table 1.

## Data Availability

The data presented in this study are stored in the hospital registry.

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
