# Peer review of "Nomogram Predicting In-Hospital Mortality in Patients with Myocardial Infarction Treated with Primary Coronary Interventions Based on Logistic and Angiographic Predictors"

_biomedicines, 2025, doi:10.3390/biomedicines13030646_

Round 1
Reviewer 1 Report
Comments and Suggestions for Authors
- Expand the introduction with more recent references and a broader discussion of existing risk models for better context.
- Clarify why specific variables were selected and explain how missing data was handled to avoid bias.
- Add clear annotations to figures like the calibration curves and decision analysis to improve readability.
- Compare the ANGIOLOG IH Risk Score with established models like GRACE.
- Discuss the implications of logistical predictors.
The manuscript would benefit from professional language editing to address these issues comprehensively
Author Response
Dear Sir/Madam
Thank you for taking the time to review our manuscript and all comments! They certainly allowed us to improve our paper. All comments have been deeply analyzed and entered into the manuscript.
In this place, point by point we would like to answer for your comments and suggestions:
Expand the introduction with more recent references and a broader discussion of existing risk models for better context.
Thank you for your valuable opinion. In accordance with the recommendations, we have expanded the introduction and added several new models for assessing the risk of in-hospital death in a patient with a myocardial infarction. It is worth mentioning that no reference scales assessing the short-term mortality risk of patients with myocardial infarction based on large databases have been published in the literature recently, which has prompted research on this topic by our team.
Clarify why specific variables were selected and explain how missing data was handled to avoid bias.
Assuming that we deal with angiographic and logistic variables, all the above variables available in the ACS GRU registry were taken into account for analysis. A package of general variables typical for most tests, such as age, gender, type of heart attack, was added to this. Virtually none of the angiographic or logistic variables available in the registry were excluded in the analysis. Variables related to medical history or laboratory test results were intentionally omitted. In the final study groups, the percentage of missing data is very low (< 0.7%). Therefore, no imputation techniques were used, and patient cases with missing data were not included in further analyses.
Add clear annotations to figures like the calibration curves and decision analysis to improve readability.
Thank you for this comment. According to suggestions, the legends under the figures have been expanded.
Compare the ANGIOLOG IH Risk Score with established models like GRACE.
Thank you for this comment. According to suggestions, a comparison of both scales was carried out.
As the GRACE scale has been the standard for assessing mortality in patients with myocardial infarction for many years, it is good practice to compare any new risk scale with the GRACE scale. Depending on the version, the GRACE score is based on a set of indicators: clinical, electrocardiographic and laboratory parameters, as well as medical history data. In fact, all subsequent models of the GRACE score are based on the same or a similar set of risk factors assessed at hospital admission. It is currently recommended by the ESC for risk stratification and to identify indications for early invasive diagnosis in patients with NSTEMI. It has been given a class IIa, level of evidence, recommendation in the ESC guidelines: B recommendation. The c-statistic of GRACE varied according to the patient subgroup selected and was > 0.8. The score was based on the analysis of data from the GRACE registry (11389 patients with MI, both STEMI and NSTEMI, enrolled between 1 April 1999 and 31 March 2001). The ANGIOLOG IH scale is constructed from angiographic and logistic variables, has a c-statistic of 0.848, and the databases on which the scale was developed are from 2019-2020 and include patients with both STEMI and NSTEMI/UA. The GRACE score is assessed at hospital admission. Risk stratification with the ANGIOLOG IH scale is performed after completion of invasive treatment.
Discuss the implications of logistical predictors.
Thank you for this comment. Logistical aspects, their importance, current application are very broadly presented in the discussion (in the subchapter logistic variables). Significant clinical aspects of logistics are presented in the Conclusions.
The manuscript would benefit from professional language editing to address these issues comprehensively.
Thank you for this comment. Before submission to the journal, we commissioned professional language editing of the manuscript at MDPI. A certificate has been attached as proof. However, if the text still requires corrections regarding the English language, please indicate the places requiring improvement and we will try to make the corrections.
Once again we would like to sincerely thank you for a very comprehensive, insightful and in many places accurate review. All of the above comments are highly valuable for the comprehensiveness of the paper and our own scientific development.
Sincerely yours,
Authors

Reviewer 2 Report
Comments and Suggestions for Authors
Logistic and angiographic predictors are well known associated with PCI results and were extensively described previously
I didnt see any novelty of the data presented by the authors.
Limitations
# Retrospective
#Small sample size
# Single center
Discusssion is confusing too large please modified Conclusion and shorter it.
The title of the manuscript should be change, not all patients had MI
Author Response
Dear Sir/Madam
Thank you for taking the time to review our manuscript and all comments. They certainly allowed us to improve our paper. All comments have been deeply analyzed and entered into the manuscript.
In this place, point by point we would like to answer for your comments and suggestions:
Logistic and angiographic predictors are well known associated with PCI results and were extensively described previously.
Thank you for your valuable opinion. Of course, we agree that many scientific reports have been published on logistical aspects of myocardial infarction treatment. However, many of these publications remain contradictory and do not allow clear conclusions to be drawn, and very often concern only one subgroup of myocardial infarction (STEMI) or date from several years ago. The most common parameter analysed was the time of hospital admission. Our study also investigated the timing of primary coronary angioplasty (so far rarely analysed in the literature, or analysed only in the context of patients with STEMI, where it was virtually synonymous with the time of hospital admission). Another advantage associated with our manuscript is, first and foremost, the timeliness of the data, as well as the broad spectrum of patients (also very often concerning patients with NSTEMI or unstable angina).
I didn’t see any novelty of the data presented by the authors.
Thank you for this comment. One of the biggest novelties of this manuscript is the fact that logical variables are used in algorithms assessing the risk of in-hospital death of a patient with a myocardial infarction. To the best of the authors' knowledge, there is no clinical risk scale that incorporates logistic aspects into the algorithm for assessment of in-hospital mortality in patients after PCIs in a wide range of MIs. The developed risk assessment model provides more interdisciplinary and realistic insight into the assessment of the short-term prognosis of patients with MI treated with PCIs. Logistic factors, unlike other classical clinical variables, can be actively modified in specific cases and, so, the described risk model also allows a certain degree of active management of the risk of in-hospital death of patients with MI.
Limitations
# Retrospective #Small sample size # Single center
Thank you for your opinion. We have noted all of these limitations and set them out in detail in the Limitations subsection. This study is a pilot study, another study (multicentre, prospective, with more patients) is in preparation.
Discussion is confusing too large please modified Conclusion and shorter it.
Thank you very much for your comment, however, we find it ambiguous. In the first part it refers to the Discussion chapter, while in the second part it refers to the Conclusion. We have therefore decided to analyse the length of both chapters. The Discussion contains 87 lines (after excluding the subsection Limitations only 76). The Conclusion contains 17 lines. The work as a whole contains 366 lines. In view of the above, the discussion chapter accounts for about 20 %, while the conclusion chapter accounts for 4% of the total work. In the opinion of the authors, these chapters did not exceed in volume the generally accepted rules for the creation of scientific papers. Secondly, in the opinion of another reviewer, the discussion should have been expanded.
The title of the manuscript should be change, not all patients had MI.
Thank you very much for this valuable comment. According to the inclusion criteria in the ACS GRU register, only adult patients diagnosed with a myocardial infarction were included (according to ESC guidelines). All analyzes in this manuscript were based on patient subgroups from the ACS GRU registry. Ultimately, all patients who were diagnosed with myocardial infarction and underwent primary coronary angioplasty were included in the derivation and validation groups. Therefore, please explain your comment and provide further instructions on how to improve the title of paper.
Once again we would like to sincerely thank you for a very comprehensive, insightful and in many places accurate review. All of the above comments are highly valuable for the comprehensiveness of the paper and our own scientific development.
Sincerely yours,
Authors

Reviewer 3 Report
Comments and Suggestions for Authors
1. In Figure 1: The derivation set and validation set at the last squares showed N=312 and N=204 respectively, but it didn’t show how many death(in hospital death) remained after excluding missing data…ect. I think it’s important data and the distribution of In hospital death could be a bias if the percentage is very different.
2. Table 1 also should be the final remaining patients/cases, not the data before excluding missing data. Perhaps you should do all the exclusion criteria then do the random allocation to the two sets.
3. Table 2, usually univariate analysis showed us promising variables(most of the P<0.05), then we put these variables to multivariable analysis to further select significant variables. But some univariate variables P<0.05 were not used, and some variable P>0.05 were put into the multivariate analysis, why?
4. Figure 2: I thought 0.7<C-statistics<0.8 is acceptable level, the higher C-statistics value the better. So the validation set has C-statistics 0.64 is not very satisfactory? And the derivation set has C-statistics 0.84 is good, but the difference between derivation set and validation set means over fitting. And that mean the generalization ability of the model maybe not satisfactory. You didn’t mention the software you use to do the binary logistic regression. Perhaps you can try python or R so that you can adjust hyperparameters to improve the generalization ability?
Author Response
Dear Sir/Madam,
First of all, the authors would like to thank you for your thorough analysis of the article we sent. All your comments and interesting observations were appreciated and implemented by us to our paper. They certainly allowed us to improve value of this manuscript. Below we present the corrections made in accordance with your suggestions. We have tried to comply with all your comments.
In this place, point by point we would like to answer for your comments and suggestions:
In Figure 1: The derivation set and validation set at the last squares showed N=312 and N=204 respectively, but it didn’t show how many death(in hospital death) remained after excluding missing data… ect. I think it’s important data and the distribution of In hospital death could be a bias if the percentage is very different.
Thank you for your valuable opinions. Following your recommendations, we have amended Figure 1 by inserting the number of deaths in the boxes with the final number of patients in the group. The percentage of deaths in the final 2 study sets did not differ significantly between them (5,83% vs. 6,09%).
Table 1 also should be the final remaining patients/cases, not the data before excluding missing data. Perhaps you should do all the exclusion criteria then do the random allocation to the two sets.
Thank you for this comment. We have created a new table with the characteristics of the two sets: derivation and validation, including the final number of patients in each set. We have included this new table in the supplementary material under number 3.
Table 2, usually univariate analysis showed us promising variables (most of the P<0.05), then we put these variables to multivariable analysis to further select significant variables. But some univariate variables P<0.05 were not used, and some variable P>0.05 were put into the multivariate analysis, why?
Thank you for your opinion. The method of creating a multivariate analysis model mentioned above is a simplified method. In order to better reflect the reality of the model, and due to the relatively small total number of variables in our model, we initially introduced all variables into the multivariate analysis, and then step-by-step backward eliminated individual variables, each time testing the statistical significance of the resulting model. Such a way of creating a multivariate analysis is admittedly more labour-intensive, but it better reflects reality by not excluding initially many variables that are not significant for the univariate analysis (which does not exclude obtaining significance for these variables during the joint analysis with other variables, which reflects the idea of multivariate analysis).
Figure 2: I thought 0.7<C-statistics<0.8 is acceptable level, the higher C-statistics value the better. So the validation set has C-statistics 0.64 is not very satisfactory? And the derivation set has C-statistics 0.84 is good, but the difference between derivation set and validation set means over fitting. And that mean the generalization ability of the model maybe not satisfactory. You didn’t mention the software you use to do the binary logistic regression. Perhaps you can try python or R so that you can adjust hyperparameters to improve the generalization ability?
Thank you very much for this extremely valuable observation. You are absolutely right about the importance of the value of the c - statistics. The satisfaction rating we gave for our validation model was based on obtaining statistic significance for the validation model. We are aware of the limitations of this study: small number of cases, one research centre, only internal validation of the obtained model. All of them can contribute to overfitting the model. Due to the above limitations, this manuscript should be treated as a pilot study aimed at showing the potential for further research in this direction. Considering this paper in these conditions, the results obtained seem to be quite promising. Further research on logistic issues has been planned based on a larger, international pool of research centers and, therefore, a much larger number of patients. We used the Stata ver 17 for statistical analyses.
Once again, we would like to sincerely thank you for a very comprehensive, insightful and in many places accurate review. All of the above comments are highly valuable for the comprehensiveness of the paper and our own scientific development.
Sincerely yours,
Authors

Round 2
Reviewer 3 Report
Comments and Suggestions for Authors
nil
Comments on the Quality of English Languagenil